# Using Modified-Intake Plasma-Enhanced Metal–Organic Chemical Vapor Deposition System to Grow Gallium Doped Zinc Oxide

**DOI:** 10.3390/mi12121590

**Published:** 2021-12-20

**Authors:** Po-Hsun Lei, Jia-Jan Chen, Ming-Hsiu Song, Yuan-Yu Zhan, Zong-Lin Jiang

**Affiliations:** Institute of Electro-Optical and Material Science, National Formosa University, No. 64, Wunhua Rd., Huwei, Yunlin County 632, Taiwan; 10676132@gm.nfu.edu.tw (J.-J.C.); 10776132@gm.nfu.edu.tw (M.-H.S.); wmocjfdk@gmail.com (Y.-Y.Z.); 10576106@gm.nfu.edu.tw (Z.-L.J.)

**Keywords:** Ga-doped ZnO (GZO), modified-intake plasma-enhanced metal–organic chemical vapor deposition (MIPEMOCVD), n-ZnO/p-GaN light-emitting diode (LED)

## Abstract

We have used a modified-intake plasma-enhanced metal–organic chemical vapor deposition (MIPEMOCVD) system to fabricate gallium-doped zinc oxide (GZO) thin films with varied Ga content. The MIPEMOCVD system contains a modified intake system of a mixed tank and a spraying terminal to deliver the metal–organic (MO) precursors and a radio-frequency (RF) system parallel to the substrate normal, which can achieve a uniform distribution of organic precursors in the reaction chamber and reduce the bombardment damage. We examined the substitute and interstitial mechanisms of Ga atoms in zinc oxide (ZnO) matrix in MIPEMOCVD-grown GZO thin films through crystalline analyses and Hall measurements. The optimal Ga content of MIPEMOCVD-grown GZO thin film is 3.01 at%, which shows the highest conductivity and transmittance. Finally, the optimal MIPEMOCVD-grown GZO thin film was applied to n-ZnO/p-GaN LED as a window layer. As compared with the indium–tin–oxide (ITO) window layer, the n-ZnO/p-GaN LED with the MIPEMOCVD-grown GZO window layer of the rougher surface and higher transmittance at near UV range exhibits an enhanced light output power owing to the improved light extraction efficiency (LEE).

## 1. Introduction

Zinc oxide (ZnO)-based materials have attracted increasing attention for decades because of their low cost, nontoxicity, wide direct band gap, large exciton binding energy, high optical transmittance in the visible range, and stable thermochemical properties [1,2]. These remarkable materials and optoelectronic characteristics render ZnO-based materials useful in various applications such as light-emitting diodes (LEDs), solar cells, photodetectors, gas and flame sensors, missile launch detection, photocatalysts, and transparent conductive oxides (TCOs) [3,4,5,6,7,8]. With advances in technology, the need for TCO thin films that are transparent in the visible range and have low electrical resistance for use as the window layer or electrode in optoelectronics is increasing daily. The most common TCO thin films applied in optoelectronic devices are indium-doped tin oxide (ITO) thin films because they exhibit good electrical and optical characteristics; however, they are marred by the drawbacks of high cost and poor thermal properties. Pure ZnO thin films do not satisfy the requirements of a TCO; nevertheless, doped ZnO thin films show promise as an alternative to ITO. Studies have reported that In-doped ZnO thin films showed a significant reduction of oxygen vacancies (V_O_) after the incorporation of indium in the ZnO lattice at substitute sites, resulting in an increased optical gap and a high carrier concentration [9,10]. However, indium is chemically unstable in reducing ambiance, and its limited availability may make it difficult to satisfy future demands. Al-doped ZnO is an attractively and widely investigated TCO material because of its low cost, good electrical characteristics, abundance, and thermal stability [11,12,13,14]. Nevertheless, acceptor defects such as interstitial oxygen and Zn vacancies strongly compensate for the doping-induced free charge in Al-doped ZnO (AZO), leading to poor reliability [15]. GZO has attracted interest in TCO and optoelectronic applications because Ga atom has less reactivity with oxygen and is more moisture-resistant as compared with Al atom [16]. In a B-doped ZnO thin film, B acts as a substitute for the Zn site and forms a strong B-O bond, thereby suppressing oxygen vacancy formation and improving electrical performance. Additionally, B has high Lewis acid strength and might help in polarizing the electron cloud away from the oxygen anion, thereby alleviating carrier scattering to increase carrier mobility [17,18]. However, B-doped ZnO, a semiconductor-like TCO, has higher resistivity and lower carrier concentration than does a metal-like TCO such as Ga-doped ZnO (GZO). Sn can also be used as a dopant in ZnO, where the Zn^2+^ ion is replaced by the Sn^4+^ ion, to increase the carrier concentration [19,20]. Compared with other TCOs, Sn-doped ZnO exhibits reduced conductivity owing to the high mean barrier height depending on disorder of grain boundary [21]. The ionic radii of Ga (0.062 nm) and Zn (0.074 nm) and covalent bond length of Ga-O (1.92 Å) and Zn-O (1.97 Å) are similar, resulting in low deformation of the ZnO lattice even at a high doping concentration [7,22,23,24]. GZO thin films have been fabricated successfully through several deposition technologies such as molecular beam epitaxy (MBE) [25,26], pulsed laser deposition (PLD) [27,28], metal–organic chemical vapor deposition (MOCVD) [29], atomic layer deposition (ALD) [22], sputtering [30,31], thermal oxidation [32], aqueous solution deposition [33], and the sol-gel method [34]. PLD and MBE grown-GZO thin films exhibit excellent crystalline structures and low resistivity; nevertheless, scaling them up to standard industrial substrate sizes is difficult [22]. ALD, sputtering, and thermal oxidation can produce high-quality GZO thin films on a large wafer size; however, these technologies have a low deposition rate that cannot satisfy the demands of mass production. The aqueous solution deposition and the sol–gel method can be used to deposit GZO thin films on large-sized substrates; however, the crystalline structure and electric properties of such GZO thin films grown by these technologies are inferior owing to multiple crystalline orientations and grain boundary scattering. MOCVD grown-GZO thin films show a high crystalline structure, low resistivity, and reasonable growth rate; however, high-quality GZO thin films should be deposited in a high growth temperature. The plasma-enhanced MOCVD (PEMOCVD) system, which contained a radio-frequency (RF) power to assist in decomposing precursors, was used to reduce the growth temperature. Recently, ZnO-based pn junctions have emerged as a candidate for near-ultraviolet (near-UV) LEDs because of their various advantages, such as wide band gap, high exciton binding energy of 60 meV at room temperature, the ability to resist radiation damage, and cost-competitive fabrication procedures, which used plasma system, wet chemical etching process (without complicated dry-etching system), and adoption of the simple device structure. Highly crystalline ZnO is an intrinsic n-type semiconductor; however, fabricating a stable and reproducible p-type ZnO material is difficult [35,36,37]. To construct a pn junction with n-ZnO, scholars have considered p-GaN with the same crystal structure (hexagonal wurtzite) and small lattice mismatch (~1.8%) with respect to ZnO as a promising material for realizing n-ZnO/p-GaN UV LEDs [38,39].

The conventional PEMOCVD system contains an RF plasma system of parallel electrodes and a gas delivery system that the deposited films might suffer bombardment damage and non-uniform content. In this study, we grew GZO thin films with low resistivity and high transmittance at low deposition temperature by using PEMOCVD with a RF plasma system composed of electrodes parallel to the substrate surface normal and a modified intake system of a mixed tank (outside the reaction chamber) with spraying terminal (in the reaction chamber) for organic precursors; this is called modified-intake PEMOCVD (MIPEMOCVD) system. The mixed tank of organic precursor servers the function of homogeneous vapor phase organic precursors mixing while the spraying terminal takes the advantage of the uniform concentration of organic precursors (DEZn and TEGa) on a substrate. The crystalline, optical, and electric properties of MIPEMOCVD-grown GZO thin films were determined to depend on the Ga content of the thin films. Finally, we applied the optimal MIPEMOCVD-grown GZO thin film to an n-ZnO/p-GaN LED as a TCO layer and compared it with the ITO TCO layer.

## 2. Materials and Methods

Figure 1a represents a schematic MIPEMOCVD system (left-hand) and modified intake system of organic precursors (right-hand), and Figure 1b exhibits the Ga content and carrier concentration of GZO thin films grown by MIPEMOCVD and PEMOCVD systems (without modified intake system) under the different TEGa flow rate. The MIPEMOCVD system includes an RF system with the electrodes parallel to the substrate surface normal, a gas delivery system of O_2_, Ar, He, and MO precursors, a heating system, a high vacuum system, and a modified intake system composed of a mixed tank with a spraying terminal for organic precursors (red dotted circle). The Ga content and carrier concentration of MIPEMOCVD-grown GZO thin films were higher than those of thin films deposited by PEMOCVD system under the same growth condition (Figure 1b). The pressure ratio of TEGa/DEZn to achieve TEGa flow rate of 6, 7, 8, and 9 sccm with DEZn flow rate of 37 sccm are 0.206, 0.214, 0.223, and 0.238. This high difference in vapor-phase pressure and flow rate reduced the TEGa concentration on the surface of the substrate during the deposition process of MIPEMOCVD-grown GZO thin films and led to low Ga content and carrier concentration in the thin films. The carrier concentration of MIPEMOCVD-grown GZO is lower than that of PEMOCVD-grown GZO under TEGa flow rate of 9 sccm owing to the strong interstitial mechanism in MIPEMOCVD-grown GZO. The modified intake system of organic precursors serve the function of a uniform mixture of DEZn and TEGa, thereby increasing the concentration of TEGa on the surface of substrate, Ga content, and carrier concentration in the MIPEMOCVD-grown GZO thin films. In addition, metal–organic precursors will not react in the modified intake system because the oxygen entered the reaction chamber directly and the mixed metal–organic precursors will go to the reaction chamber rapidly owing to the low deposition pressure. Figure 1c shows the average thickness of MIPEMOCVD- and PEMOCVD-grown GZO thin films at upper-left (UL), lower-left (LL), center, upper-right (UR), and lower-right (LR) positions of 1.5 cm × 1.5 cm glass substrate under the DEZn and TEGa flow rates of 37 and 7 sccm. The standard deviations of thickness for MIPEMOCVD- and PEMOCVD-grown GZO thin films in Figure 1c are 4.6 and 30.6, implying that the modified intake system can be used to expand the mixed precursors of DEZn and TEGa in the reaction chamber and to obtain a well-controlled deposition rate of MIPEMOCVD-grown GZO thin films over the substrate. The Ga content in MIPEMOCVD-grown GZO thin films changed with the TEGa flow rate from 6 to 9 sccm at the deposition temperature and radio-frequency (RF) power of 245 °C and 350 W, respectively. Table 1 lists the deposition parameters for the MIPEMOCVD-grown GZO thin films. The chamber pressure and deposition temperature maintain 2.25 × 10^−4^ Pa and 245 °C during the deposition process. To form a stable and uniform plasma in the chamber during GZO thin film deposition, the optimal chamber pressure was 2.25 × 10^-4^ Pa in our previous study [40]. Additionally, to obtain an optimal deposition temperature of MIPEMOCVD-grown GZO, the FWHM of (002) peak of XRD spectra, resistivity, and transmittance at 405 nm for MIPEMOCVD-grown GZO thin films grown at different deposition temperatures were shown in Table 2. With rising deposition temperature, the reactants including zinc atoms, gallium atoms, and oxygen free radicals, can move rapidly to the lowest energy site. Therefore, the FWHM of (002) peak in XRD spectra decreases with increasing deposition temperature. The MIPEMOCVD-grown GZO thin film grown at 255 °C exhibits a higher resistivity as compared with that grown at 245 °C possibly due to the low sticking coefficient of Ga atoms, leading to a low carrier concentration. The optimal deposition temperature of GZO thin film to obtain the lowest resistivity and highest transmittance at 405 nm is 245 °C.

The surface morphology of the MIPEMOCVD-grown GZO thin films on sapphire substrates was measured using scanning electron microscopy (SEM) and atomic force microscopy (AFM) instruments (D13100, Digital Instruments, Veeco Metrology Group, New York, NY, USA). The elemental content in the MIPEMOCVD-grown GZO thin film was analyzed through energy-dispersive X-ray spectroscopy (EDX). The crystalline structure of the MIPEMOCVD-grown GZO thin film was characterized by X-ray diffraction (XRD) patterns using a Bruker D8 advanced diffractometer with CuKα radiation (λ = 0.154 nm). The transmittance of the MIPEMOCVD-grown GZO thin films in the visual range was measured using a UV–visible–near-infrared (UV–Vis–NIR) spectrophotometer (UVD-350). The electrical properties, including carrier concentration, carrier mobility, and resistivity, were assessed through Hall measurements (HMS-5000) at room temperature. The deposition rate and thickness of the MIPEMOCVD-grown GZO thin films grown on the sapphire substrate were maintained at 10 nm/min and 300–350 nm for crystalline, optical, and electrical measurements. The MIPEMOCVD-grown GZO thin film and the sputter-deposited ITO thin film with a resistivity of 6.69 × 10^−4^ Ωcm were used as the window layers for n-ZnO/p-GaN LEDs.

Figure 2a,b illustrates the schematic device structures for n-ZnO/p-GaN LEDs with MIPEMOCVD-grown GZO and ITO window layers, respectively. The thicknesses of ITO, MIPEMOCVD-grown GZO, n-ZnO, and p-GaN grew on the sapphire substrate were 350, 350, 30, and 200 nm, respectively. The wafers were then patterned using a standard photolithographic process to define square mesas as the emitting regions by partially etching the exposed MIPEMOCVD-grown GZO/n-ZnO and ITO/n-ZnO. A Ti/Pt/Au alloy with a thickness of 50/50/900 nm was used as the ohmic contact metal on the MIPEMOCVD-grown GZO, ITO, and p-GaN contact regions. The wafer was then annealed in N_2_ atmosphere for 5 min at 450 °C. The size of the emission window for the InGaN/GaN LEDs with MIPEMOCVD-grown GZO and ITO was 300 × 300 μm^2^. Traditional light output intensity–current (L–I) measurements were performed using a current measurement unit and a calibrated power meter (Keithley 2520). To realize the linewidth and center wavelength of the emitted light, the emission spectra of n-ZnO/p-GaN LEDs with MIPEMOCVD-grown GZO and ITO window layers were detected using an optical spectrum analyzer (70682 NS).

## 3. Results and Discussion

To examine the effects of Ga content on the structural characteristics of the GZO thin films, we have investigated the Zn and O contents of ZnO thin film and Ga, Zn, and O contents of MIPEMOCVD-grown GZO thin films from the spectra of EDX measurement shown in Figure 3. The x-axis and y-axis of EDX are X-ray energy and count of elements. The MIPEMOCVD-grown GZO thin films with varied Ga content of 1.59 to 5.42 at% can be obtained by adjusting the flow rate and temperature of the TEGa source.

The XRD spectra for ZnO and MIPEMOCVD-grown GZO thin films with varying Ga content were shown in Figure 4a. The schematic structures for ZnO, MIPEMOCVD-grown GZO thin films with substituted Ga atoms (Ga content of 1.95 and 3.01 at%) and interstitial Ga atoms (4.52 and 5.42 at%)) were shown in Figure 4b–d. Table 3 exhibits the 2θ angle and full width at half maximum (FWHM) of (002) preferential peak in XRD spectra and the calculated grain size for ZnO thin film and MIPEMOCVD-grown GZO thin films with varying Ga content. According to the XRD patterns (Figure 4), all thin films including ZnO and the MIPEMOCVD-grown GZO thin films with varying Ga content represented a polycrystalline wurtzite structure and exhibited an extremely pronounced (002) orientation that was perpendicular to the substrate. The Ga atoms in ZnO thin film would be substituted or interstitial because the ionic radius of the Ga atom is smaller than that of the Zn atom. The substituted or interstitial Ga atoms in the MIPEMOCVD-grown GZO thin film created defects owing to the strain resulting from the different bond lengths of Ga-O and Zn-O. These defects distorted the local order of the crystal structure of the MIPEMOCVD-grown GZO thin films and then led to a decrease in the (002) peak intensity as compared with ZnO thin film [2]. The (100) and (101) peaks in Figure 4 could be attributed to the low deposition temperature [22]. Because the crystalline structure of the ZnO thin films was distorted owing to the doping of Ga atoms, the intensities of the (100) and (101) peaks for the MIPEMOCVD-grown GZO thin films were more obvious than those of the peaks for the intrinsic ZnO thin film. Because Ga atoms were homogeneously incorporated into the ZnO crystal structure, the diffraction angles of (002) peak for MIPEMOCVD-grown GZO thin films with Ga contents of 1.95 and 3.01 at% were close to that of ZnO thin film in the inset of Figure 4. As the Ga contents of MIPEMOCVD-grown GZO thin films rose to 4.52 and 5.42 at%, the diffraction angles of (002) peak shifted to the small angle as compared with that of ZnO thin film in the inset of Figure 4. The diffraction angle of (002) peak for high Ga substitution at the Zn site in GZO is larger than that for ZnO because the bond length of Zn-O is larger than that of Ga-O. According to the Scherrer formula [41], the crystalline size of GZO will decrease with increasing Ga content owing to the larger diffraction angle. However, MIPEMOCVD-grown GZO thin films with Ga contents of 4.52 and 5.42 at% exhibit a larger calculated grain size and smaller diffraction angles of (002) peak as compared with ZnO thin film (Table 3), which are attributed to the stronger interstitial mechanism in MIPEMOCVD-grown GZO thin films with Ga content above 4.52 at% [42].

Figure 5a–e depicts SEM images of ZnO and MIPEMOCVD-grown GZO thin films with 1.95, 3.01, 4.52, and 5.42 at% Ga grown at a deposition temperature and RF power of 245 °C and 350 W, respectively. These images reveal pillar-like grains with a hexagonal wurtzite structure [43]. The MIPEMOCVD-grown GZO thin films grown under a low deposition pressure and low growth temperature are ordered and directional because the deposition rates of bulk and initial stage are slow and constant for the low deposition pressure but high and slow for the high deposition pressure. The high deposition pressure leads to a porous initial incubation layer in the beginning of the growth for GZO thin film [44]. The grain size (Figure 5a–e) of MIPEMOCVD-grown GZO thin films increased with the Ga content, matching the tendency indicated in Table 3. Highly doped Ga atoms, as well as Al atoms doped in ZnO films, served as a surfactant for grain growth; consequently, the grain size of the MIPEMOCVD-grown GZO thin films increased with Ga content [45]. Additionally, the FWHM of (002) peak for MIPEMOCVD-grown GZO thin films decreases with increasing the Ga content from 0 to 5.42 at% as shown in Table 3. The MIPEMOCVD-grown GZO thin film with a higher Ga content exhibits a larger grain size that reduces the lattice distortion and structural defects caused by the in-plane stress; as a result, the FWHM of (002) peak for MIPEMOCVD-grown GZO thin film with high Ga content becomes narrow [46].

To analyze the electrical properties of the MIPEMOCVD-grown GZO thin films with varying Ga content, Hall measurements were conducted at room temperature. Figure 6 illustrates the resistivity, carrier mobility, and carrier concentration of the ZnO and MIPEMOCVD-grown GZO thin films as a function of Ga content. The intrinsic ZnO thin film had a carrier concentration of 4.51 × 10^18^ cm^−3^ and a resistivity of 2.32 Ωcm. As the Ga content increased to 3.01 at%, the carrier concentration of the GZO film increased and reached a maximum value of 4.81 × 10^21^ cm^−3^; the resistivity of the film decreased and reached a minimum value of 8.36 × 10^−4^ Ωcm. The diffraction angles of (002) peak for MIPEMOCVD-grown GZO thin films with Ga contents of 1.95 and 3.01 at% are close to that for ZnO as compared with MIPEMOCVD-grown GZO thin films with Ga contents of 4.52 and 5.42 at%, implying a strong substituted mechanism rather than an interstitial mechanism for MIPEMOCVD-grown GZO thin films with Ga contents of 1.95 and 3.01 at%. The ratio of Zn/O is 0.76 and 0.65 for MIPEMOCVD-grown GZO thin films with Ga contents of 1.95 and 3.01 at%, respectively. The 3.01 at% Ga-contented GZO thin film with a lower Zn/O ratio implies more Ga substitution to Zn-site in GZO as compared to 1.95 at% Ga-contented GZO, leading to a high carrier concentration. However, further increasing the Ga content to 4.52 and 5.42 at% will reduce the carrier concentration (Figure 6) owing to the strong interstitial mechanism. The interstitial mechanism leads to fewer electrons released from Ga atoms; therefore, the carrier concentration decreases with increasing Ga content [16,41]. The carrier mobility of the MIPEMOCVD-grown GZO thin films is influenced by multiple scattering mechanisms, which can be combined as follows [47]:(1)$$\frac{1}{\mu }=\frac{1}{\mu_{i}}+\frac{1}{\mu_{\mathrm{gb}}}+\frac{1}{\mu_{p}},$$
where μ_i_, μ_gb_, and μ_p_ are the mobility caused by ionized impurity scattering, grain boundary scattering, and phonon scattering, respectively. The μ_p_ value is inversely proportional to the ambient temperature and is dominant only at high temperatures; therefore, the carrier mobility μ_p_ could be considered negligible in this investigation because the measurements were performed at room temperature. This study focused on the carrier mobility of ionized impurities and grain boundary scattering. Based on the highly degenerated electron gas model at a high carrier concentration, the mean free path of the free carrier (λ), which determines the carrier mobility, can be expressed as follows [48]:(2)λ=3π213he-2ρn-23
where h is Planck’s constant, ρ is the electrical resistivity, and n is the carrier concentration (10^20^ cm^−3^). The mean free paths in the MIPEMOCVD-grown GZO thin films with 1.95, 3.01, 4.52, and 5.42 at% Ga were 7.0, 33.4, 14.3, and 3.7 nm, respectively. The mean free path of the carrier was shorter than that of the grain size presented in Table 3; thus, besides ionized impurity scattering, the effect of grain boundary scattering should be considered for the carrier mobility. In summary, the ionized-impurity scattering and the grain-boundary scattering mechanisms would be considered for carrier mobility of the MIPEMOCVD-grown GZO thin films. Because the grain size increased with the Ga content in the MIPEMOCVD-grown GZO thin films (Table 3 and Figure 5), large-sized grains alleviated grain boundary scattering; thus, the mobility of the MIPEMOCVD-grown GZO thin films increased as the Ga content increased from 0 to 3.01 at%. However, the MIPEMOCVD-grown GZO thin films with 4.52 and 5.42 at% Ga exhibited a larger grain size but lower mobility compared with those for which the Ga content was less than 4.52 at%. As Ga content increases to 4.52 and 5.42 at%, the interstitial mechanism [42], which might increase the probability of carrier scattered with interstitial Ga atoms, determines the site of Ga atoms in the ZnO crystal structure; therefore, the carrier mobility for the MIPEMOCVD-grown GZO thin film with Ga content above 4.52 at% is low.

Figure 7 displays the transmittance of ZnO, ITO, and MIPEMOCVD-grown GZO thin films with varying Ga content over the wavelength range of 300–850 nm and the inset of Figure 7 presents the average transmittance of the MIPEMOCVD-grown GZO thin films in the visible range. The thickness of all thin films maintained at 300–350 nm. The spectra fluctuated because of the multiple reflections at the interface [31]. All MIPEMOCVD-grown GZO thin films grown with different Ga contents were fairly smooth and exhibited a high average transmission of 89.9–92% in the visible range (Figure 7), indicating their low surface roughness and homogeneous crystalline structure. Moreover, the average transmittance of the MIPEMOCVD-grown GZO thin films increased as the Ga content increased from 0% to 3.01 at% and then decreased when the Ga content was 4.52 and 5.42 at%. The increased average transmittance might be attributed to the enlarged grain size and highly crystalline structure (low FWHM in XRD spectra), which reduced the reflection and scattering of incident light. The decrease in the average transmittance of the MIPEMOCVD-grown GZO thin films when the Ga content was 4.52 and 5.42 at% can be attributed to the degradation of crystal quality and scattering of interstitial Ga atoms.

Finally, we applied the ITO and optimal MIPEMOCVD-grown GZO (3.01 at% Ga) thin films to n-ZnO/p-GaN LEDs as window layers and measured the optoelectronic properties of these devices. Figure 8a illustrates the light output power as a function of the injection current for the n-ZnO/p-GaN LEDs with the optimal MIPEMOCVD-grown GZO (3.01 at% Ga) and ITO window layers, and its inset depicts AFM images of the optimal MIPEMOCVD-grown GZO and ITO thin films grown on a sapphire substrate. The device-to-device standard deviations of light output power under the driving current of 20 mA for n-ZnO/p-GaN LEDs with ITO and optimal MIPEMOCVD-grown GZO window layer are approximately 0.61 and 0.42. The n-ZnO/p-GaN LED with the optimal MIPEMOCVD-grown GZO exhibited a higher light output power than the device with ITO at the same injection current owing to the higher external quantum efficiency (EQE) as shown in Figure 8a. The EQE defined as the ratio of the number of emitting photons to injected electrons is the product of the internal quantum efficiency (IQE) and light extraction efficiency (LEE), which determine the light output power. The IQE levels observed for the n-ZnO/p-GaN LEDs the optimal MIPEMOCVD-grown GZO and ITO window layers were equal because of the similar device structures. Consequently, the EQE levels of n-ZnO/p-GaN LEDs for the optimal MIPEMOCVD-grown GZO and ITO window layers depend on the LEE. Light scattering should be realized at the interface between the window layer and air because the LEE of an LED with a textured window layer depends on the intensity of scattering light. The normalized intensity of scattering light can be expressed as follows [49]:(3)$$I\left(\lambda ,\varphi \right)=\frac{A_{\mathrm{OMS}}}{A}\cos\varphi {\left|\frac{k^{2}\left(n^{2}-1\right)}{4\pi r}\iint_{A}\left[\frac{1}{\mathrm{iK}}\left(1-e^{-\mathrm{ikh}}\right)\right]e^{-i\left(K_{x}x+K_{y}y\right)}\mathrm{dxdy}\right|}^{2}$$
where A_OMS_ is the area of the optical measurement system; A is the area of the textured window layer; φ is the scattering angle; n is the refractive index; K = $k\sqrt{1-s_{x}-s_{y}}-1$; s_x_ and s_y_ are the unit vectors in the x and y directions, respectively; h is the height function related to the surface morphology of the textured window layer, and K_x_ and K_y_ satisfy K_x_^2^ + K_y_^2^ = (ksinφ)^2^. Assuming the same area of the optical measurement system, similar refractive indices for ITO and MIPEMOCVD-grown GZO, and small kh and φ approximations (cosφ ≈ 1 and sinkh ≈ kh), the normalized intensity of scattering light depends on the root mean square (RMS) roughness squared of window layer [50]. Because the RMS roughness of the optimal MIPEMOCVD-grown GZO (6.45 nm) is higher than that of ITO (5.46 nm), as shown in the inset of Figure 8a, strong light scattering occurs at the interface between the optimal MIPEMOCVD-grown GZO layer and air, explaining the higher light output power observed for the n-ZnO/p-GaN LED with the optimal MIPEMOCVD-grown GZO window layer as compared with that with the ITO window layer under the same injection current. Additionally, the electrical property of MIPEMOCVD-grown GZO thin film would be affected by the underlayer or substrate. For example, the carrier concentration and resistivity of MIPEMOCVD-grown GZO thin films are 4.81 × 10^21^ cm^−3^ and 8.35 × 10^−4^ Ωcm for sapphire and 4.13 × 10^21^ cm^−3^ and 9.21 × 10^−4^ Ωcm for glass substrate. These results imply that the n-ZnO can be used as a seed layer to reduce the resistivity of MIPEMOCVD-grown GZO thin film, bringing about an improved light output power for n-ZnO/p-GaN LED with the optimal MIPEMOCVD-grown GZO window layer. In summary, the increased light output power of n-ZnO/p-GaN LED with the optimal MIPEMOCVD-grown GZO thin film was attributed to the following factors: (1) the higher transmittance at emitting wavelength, (2) rougher GZO surface to result in a strong light scattering between air and GZO, and (3) lower specific contact between the interface of MIPEMOCVD-grown GZO and n-ZnO. Figure 8b shows the room-temperature electroluminescence (EL) spectra for the n-ZnO/p-GaN LEDs with the optimal MIPEMOCVD-grown GZO and ITO window layers at a driving current of 20 mA, and its insets show micrographs of light emission for the n-ZnO/p-GaN LEDs with the optimal MIPEMOCVD-grown GZO and ITO window layers. The EL spectra of n-ZnO/p-GaN LEDs with the optimal MIPEMOCVD-grown GZO and ITO window layers indicated FWHM of 30 and 34 nm and center peaks located at 405 and 404 nm (Figure 8b). The n-ZnO/p-GaN LED with the optimal MIPEMOCVD-grown GZO window layer represents a narrower FWHM and higher intensity than that with the ITO window layer in Figure 8b due to the high light output power and LEE. The light output power and LEE of an LED device can be affected by the transmittance of the window layer. ITO window layer exhibits a lower transmittance in UV region as compared with MIPEMOCVD-grown GZO window layer (black line in Figure 7) owing to the strong absorption in UV region [38,51]. The carrier concentrations of n-ZnO and p-GaN were about 10^18^ cm^−3^; therefore, the width of the depletion region in n-ZnO and p-GaN is almost equal. The holes from p-GaN and electrons from n-ZnO are injected into n-ZnO and p-GaN to undergo radiative recombination; this leads to multiple emissions including near-band-edge emission from n-ZnO, radiative recombination at the interface of n-ZnO/p-GaN, and Mg-acceptor-related emission from p-GaN, as a result, a center wavelength located at 405 and 404 nm in was found for the emission spectra of n-ZnO/p-GaN LED with the optimal MIPEMOCVD-grown GZO and ITO window layers [38,52]. Additionally, the center wavelength of n-ZnO/p-GaN LEDs with GZO shows a red-shift possible due to high series resistance. The high series resistance results in an increasing junction temperature, leading to a slight reduction of bandgap and red-shift in the emitting spectrum. The light intensity observed for the n-ZnO/p-GaN LED with the optimal MIPEMOCVD-grown GZO window layer was higher than that observed for the device with ITO window layer owing to the improved LEE.

## 4. Conclusions

We investigated the crystalline structure, electrical properties, and optical characteristics of GZO thin films with varying Ga content grown using an MIPEMOCVD system. The Ga atoms in the ZnO matrix exhibited a substitution behavior when the Ga content was 1.95–3.01 at%; however, they exhibited a strongly interstitial behavior when the Ga content exceeded 3.01 at% owing to the small ionic radii of Ga atom and bond length of Ga-O. The MIPEMOCVD-grown GZO thin film with 3.01 at% Ga had a higher carrier concentration (4.81 × 10^21^ cm^−3^) and lower resistivity (8.36 × 10^−4^ Ωcm) than those of the ZnO thin film (carrier concentration of 4.51 × 10^18^ cm^−3^ and resistivity of 2.32 Ωcm). All MIPEMOCVD-grown GZO thin films had an average transmittance of 89.9–92% over the wavelength range of 300–850 nm. The n-ZnO/p-GaN LED with the optimal MIPEMOCVD-grown GZO window layer exhibited a higher light output power than that with an ITO window layer because the optimal MIPEMOCVD-grown GZO window layer with a high RMS roughness and transmittance in near UV range enhances the LEE of the n-ZnO/p-GaN LED

## Figures and Tables

**Figure 1 micromachines-12-01590-f001:**
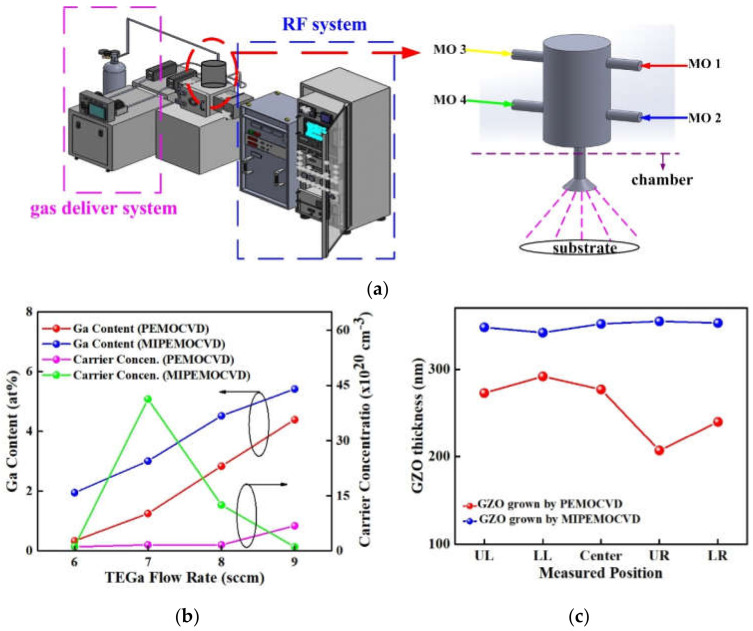
(**a**) Schematic MIPEMOCVD system and the modified intake system of organic precursors (right-hand), (**b**) Ga content and carrier concentration of PEMOCVD- and MIPEMOCVD-grown GZO thin films grown with varying TEGa flow rate, and (**c**) the average thickness of MIPEMOCVD- and PEMOCVD-grown GZO thin films at different position on glass substrate.

**Figure 2 micromachines-12-01590-f002:**
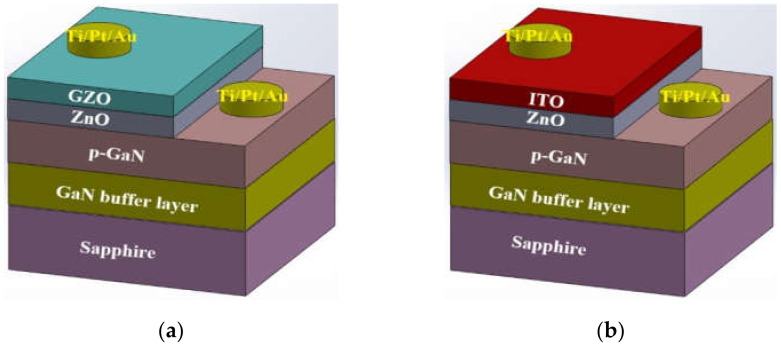
Schematic of device structures of n-ZnO/p-GaN LEDs with (**a**) MIPEMOCVD-grown GZO and (**b**) ITO.

**Figure 3 micromachines-12-01590-f003:**
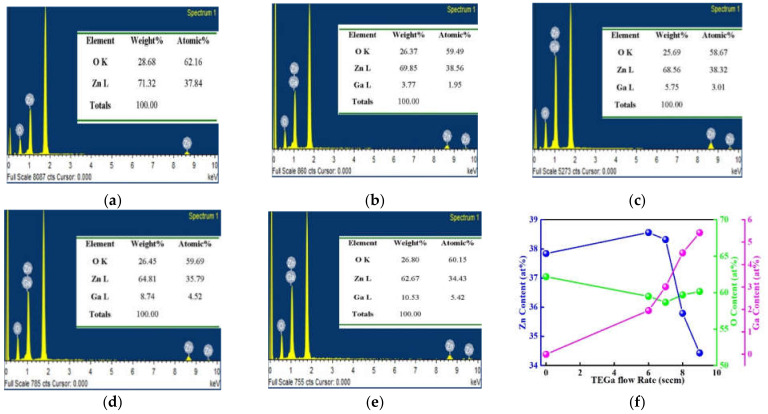
The EDX spectra of (**a**) ZnO and MIPEMOCVD-grown GZO thin films with Ga contents of (**b**) 1.95, (**c**) 3.01, (**d**) 4.52, and (**e**) 5.42 at%. (**f**) summary of Zn, O, and Ga content of ZnO and MIPEMOCVD-grown GZO with varying TEGa flow rate.

**Figure 4 micromachines-12-01590-f004:**
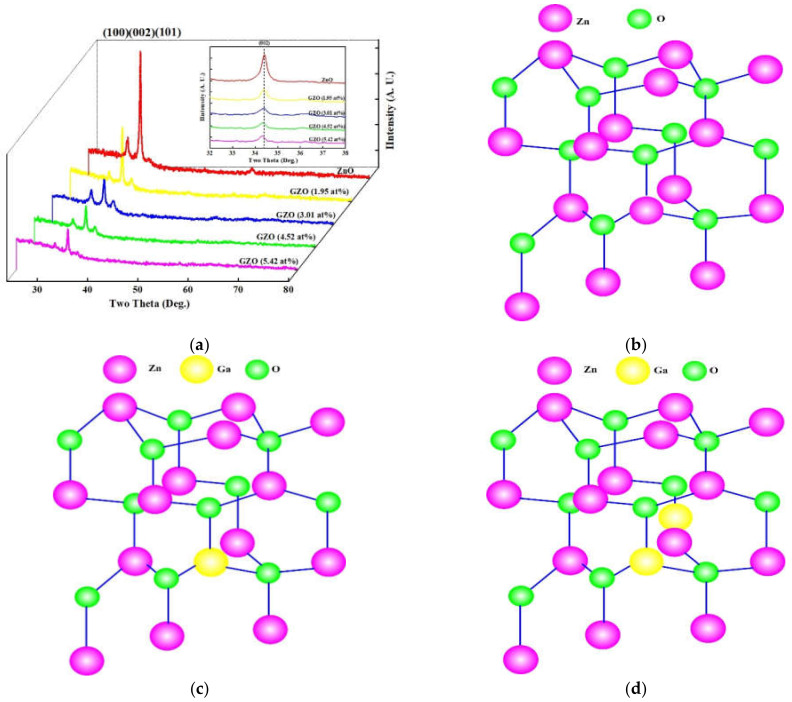
(**a**) XRD spectra of MIPEMOCVD-grown GZO thin films with varying Ga content and the schematic crystal structure for (**b**) ZnO, (**c**) GZO with substituted Ga atoms (Ga contents of 1.95 and 3.01 at%), and (**d**) GZO with interstitial Ga atoms (Ga contents of 4.52 and 5.24 at%).

**Figure 5 micromachines-12-01590-f005:**
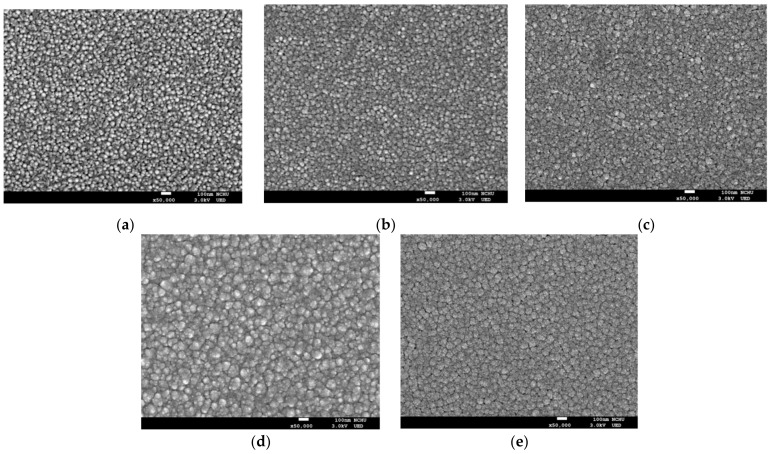
SEM images of (**a**) ZnO and MIPEMOCVD-grown GZO thin films with (**b**) 1.95, (**c**) 3.01, (**d**) 4.52, and (**e**) 5.42 at% Ga.

**Figure 6 micromachines-12-01590-f006:**
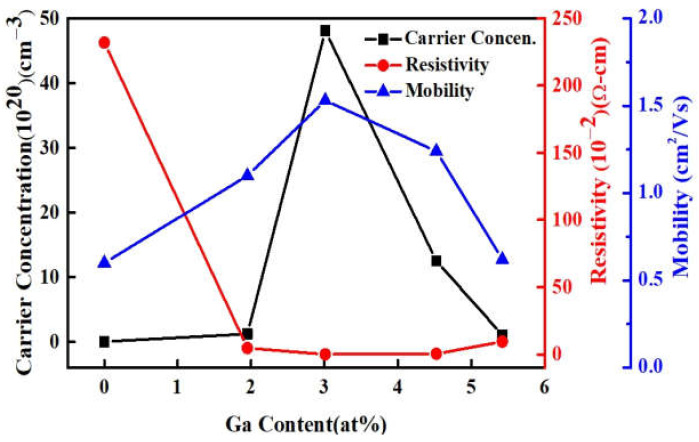
Hall measurements of ZnO and MIPEMOCVD-grown GZO thin films as a function of Ga content.

**Figure 7 micromachines-12-01590-f007:**
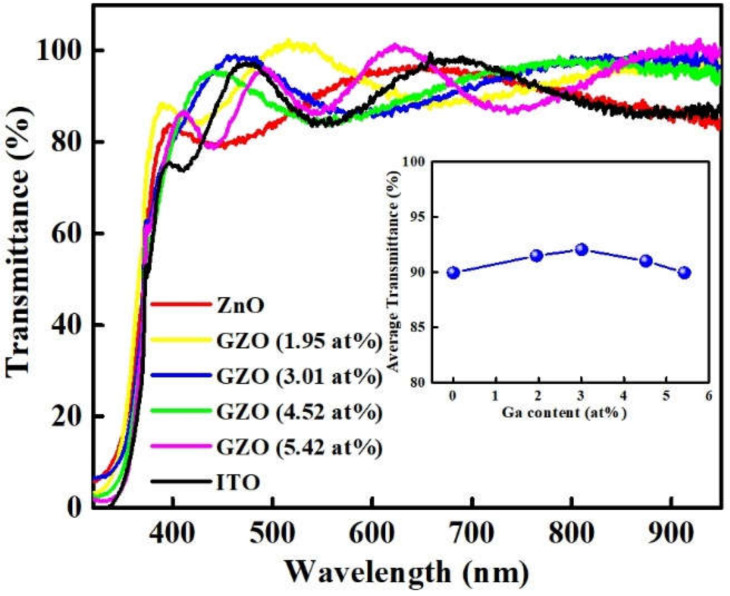
The transmittance of ZnO, ITO, and MIPEMOCVD-grown GZO thin films with varying Ga content over the wavelength range of 300–850 nm and the inset shows the average transmittance over the visible range for MIPEMOCVD-grown GZO thin films with varying Ga content.

**Figure 8 micromachines-12-01590-f008:**
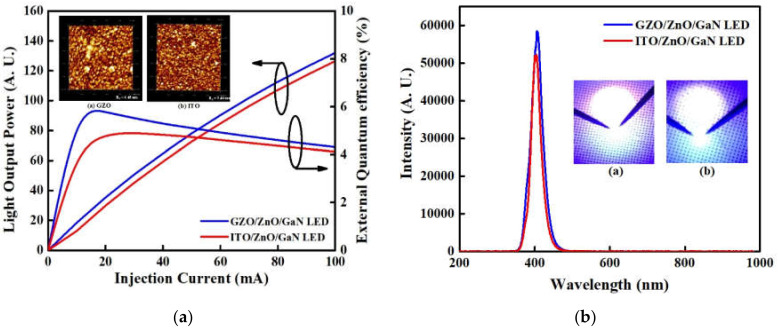
(**a**) Light output power and external quantum efficiency as a function of injection current and, (**b**) room-temperature EL spectra for n-ZnO/p-GaN LEDs with MIPEMOCVD-grown GZO and ITO window layers. The insets in (**a**) and (**b**) show the AFM images and micrographs of light emission for the n-ZnO/p-GaN LEDs with MIPEMOCVD-grown GZO and ITO window layers.

**Table 1 micromachines-12-01590-t001:** Deposition conditions of GZO thin films grown by using MIPEMOCVD system.

**Chamber pressure (Pa)**	2.25 × 10^−4^
**Deposition temperature (°C)**	245
**Deposition rate of MIPEMOCVD-grown GZO (nm/min)**	10
**RF power (W)**	350
**DEZn flow rate (sccm)**	37
**O_2_ flow rate (sccm)**	7
**TEGa flow rate (sccm)**	6–9

**Table 2 micromachines-12-01590-t002:** The 2θ angle and FWHM of (002) peak of XRD spectra, resistivity, and transmittance at 405 nm of MIPEMOCVD-grown GZO with varying deposition temperature.

Temperature (°C)	FWHM of (002) Peak (Deg.)	Resistivity (Ωcm.)	Transmittance at 405 nm (%)
185	0.339	1.48 × 10^−1^	81.92
225	0.333	3.84 × 10^−2^	83.59
235	0.329	2.11 × 10^−2^	85..35
245	0.324	8.35 × 10^−4^	86.06
255	0.318	9.53 × 10^−3^	85.76

**Table 3 micromachines-12-01590-t003:** The 2θ angle and FWHM of (002) peak of XRD spectra and calculated grain size for ZnO and MIPEMOCVD-grown GZO with varying Ga content.

Ga Content (at%)	2θ Angle of (002) Peak (Deg.)	FHWM of (002) Peak (Deg.)	Calculated Grain Size (nm)
0	34.40	0.378	44.25
1.95	34.39	0.356	46.99
3.01	34.38	0.324	51.86
4.52	34.35	0.320	52.28
5.42	34.32	0.316	52.94

## Data Availability

The data presented in this study are available in the manuscript.

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
