# Peer review of "Using Modified-Intake Plasma-Enhanced Metal–Organic Chemical Vapor Deposition System to Grow Gallium Doped Zinc Oxide"

_micromachines, 2021, doi:10.3390/mi12121590_

Round 1

Reviewer 1 Report

The manuscript reports the MOCVD preparation of Ga doped ZnO as transparent conductive thin fims, which is imprtant to current LED and display industry. Here I have several minor comments.

  1. The figures ar not of high resolution. And in Figure 3, data should be extracted and replotted for clarity instead of original raw data.
  2. In figure 7, the transmission should be related to film thickness. This should be considered in the comparison of the performace.
  3. The long-time and enviroment stability of the LED with Ga:ZnO film should be test since the motivation is to compete with industrial usage of ITO.
  4. Cystal structure for Ga:ZnO at different Ga concentrations should be presented.

Author Response

Reply to reviewers’ comments

In the comments made by the reviewer, some technical and writing comments made from the reviewer for this paper are useful. We thank the constructive comments and made the following revisions. The specific comments made by the reviewers and rebuttals are listed as following.

Reviewer 1

  1. The figures ar not of high resolution. And in Figure 3, data should be extracted and replotted for clarity instead of original raw data.

We have enhanced the resolution of the figures in the manuscript and added Figure 3 (e) to show the raw data from EDX spectra in page 6.

  1. In figure 7, the transmission should be related to film thickness. This should be considered in the comparison of the performance.

We thank reviewer`s suggestion. The thickness of all thin films maintained 300-350 nm, and we have added it in page 10 in bold green.

  1. The long-time and enviroment stability of the LED with Ga:ZnO film should be test since the motivation is to compete with industrial usage of ITO.

The resistivity of GZO thin films will degrade from 8.36 × 10−4 to 10-3 Ωcm for 60 days in the environment of 40% humidity and room temperature. To maintain a stable LED performance, the LED chip composed of GZO window layer should be packed as well as industrial usage of ITO.

  1. Crystal structure for Ga:ZnO at different Ga concentrations should be presented.

We have added the schematic crystal structures of ZnO, GZO with interstitial Ga atoms (Ga contents of 1.95 and 3.01 at%), and GZO with substituted Ga atoms(Ga contents of 4.52 and 5.24 at%) in Fig. 4 (b), (c), and (d). The description of Fig. 4 was shown in page 6 in bold green.

Reviewer 2 Report

  1. Abstract of the review:
  • In the manuscript, the authors introduce a modified plasma-enhanced metal-organic chemical vapor deposition technique to grow zinc oxide uniformly doped with gallium. This technique is interesting as it shows potential to grow materials with uniform doping and have optical, electrical, and structural characteristics suitable for optoelectronic applications. Hence, the primary topic of the manuscript is potentially considerable for publication.
  • The authors perform sufficient characterizations to convey their approach and explain the results. However, there are several gaps in the explanation of the growth technique and interpretation of the data. The incorporation of Ga in ZnO is not clear. Application-related impacts of the results are not clear. This hinders a clear understanding of the manuscript. Recommended improvements are elaborated in Section III of this document.
  • The manuscript has multiple language-related and sentence-formation errors. Multiple paragraphs in the manuscript are very long (as long as a page). These should be addressed.

  1. Reviewer decision: Reconsider after major revision

III. Reviewer comments:

(Line numbers in the following refer to the line numbers that appear in a column in the blank space besides the manuscript text towards the right.)

  1. On line 57, authors mention that GZO has less reactivity in oxygen and lesser moisture resistance compared with other dopants. It would be helpful to elaborate more on how these properties would make GZO interesting for optoelectronics.
  2. On line 72, more elaboration on the differences in crystalline structure and electrical properties of GZO thin films using different growth technologies will be helpful. Elaboration on applications where these properties are inferior when grown by other technologies will also be helpful.
  3. Labeling of components in the Figures 1 (a) and 1(b) will enable better understanding of the schematics.
  4. Caption in Figure 1 (c) is inaccurate and needs to be updated
  5. The carrier concentration at Ga content of 9% is lower when grown using MIPEMOCVD than PEMOCVD. Elaboration on this trend would be appreciated.
  6. In the modified intake system, the organic precursors are mixed in a tank outside the reaction chamber. For MOCVD, the reactions occur in the reaction chamber under specific growth conditions (temperature, pressure, etc.). How is it made sure that reactions do not occur in the tank outside the reaction chamber before the precursors are sprayed in the reaction chamber? Elaboration on this will be appreciated here and in the manuscript.
  7. In caption 1 (c), authors mention that the figure represents thickness of GZO thin films. However, it appears that thickness of only one sample is presented in the figure. If this one sample is representative of all other grown samples, please mention so and update the caption.
  8. Selection of chamber pressure and deposition temperature for the growth should be justified. Are these optimum conditions for ZnO growth? Please mention the basis for the growth conditions.
  9. On line 164, do the authors mean “annealed” instead of “alloyed”? If so, please correct the error.
  10. On line 168, do the authors mean “emitted” light instead of “emitting light”? If so, please correct the error.
  11. Please clearly mention the x-axis and y-axis titles on the plots in Figure 3.
  12. Please mention or explain if the increase in Ga content in Figure 3 was proportional (or directly related) to the input TEGa/DEZn ratio.
  13. Line 210 makes an interesting point regarding the incorporation mechanism of Ga in ZnO. It would be great if the sentence could be split into 2-3 sentences or updated to convey the message clearly.
  14. Line 238 seems to have an error. The word (high) after deposition seems erroneous. Please correct or clarify the sentence.
  15. On line 242, authors mention Al atoms doped in   Is this result from this manuscript or from literature? Please clarify.
  16. The authors state that reduction in FWHM of  (002) peak indicate formation of structural defects (line 248). Please provide more elaboration on this.
  17. Structural characterization (XRD) indicated that Ga occupied more interstitial sites in ZnO (even in GZO with 1.95 and 3.01% Ga). As per line 261, authors indicate that Ga substituted Zn more than occupying interstitial sites. Please explain these observations.
  18. On line 284, authors mention that the trend of carrier mean free path is comparable to grain size. The grain size increased proportional to the Ga content. But the carrier mean free path does not show such a trend. Please elaborate and update the corresponding sentence(s) in the manuscript.
  19. In legend of Figure 6, do authors mean “carrier density” instead of “carrier”? If yes, please update legend in Figure 6.
  20. In line 379, authors mention that the p-n heterojunction’s depletion region is “fairly”. Please elaborate on this.
  21. Referring to line 387, please elaborate the effect of joule-heating on the center wavelength of emission spectra. If the effect is not significant, please mention so.
  22. The light output power from GZO/ZnO/GaN lED was more than ITO/ZnO/GaN LED, but the EQE was similar. The authors provide explanation on this around line 334. Please show plots of the EQE. Which optoelectronic applications are influenced more by EQE which are influenced more by light output power characteristics? Please elaborate on this and more on the relevance and application-related impact of results from Figure 8.

Author Response

Reply to reviewers’ comments

In the comments made by the reviewer, some technical and writing comments made from the reviewer for this paper are useful. We thank the constructive comments and made the following revisions. The specific comments made by the reviewers and rebuttals are listed as following.

Reviewer 2

  1. The manuscript has multiple language-related and sentence-formation errors. Multiple paragraphs in the manuscript are very long (as long as a page). These should be addressed.

Our manuscript has been edited by the English-speaking colleague (Wallace academic editing, Taiwan). However, some sentences may be changed away from the original think and viewpoint. The typo, ambiguous sentences, and unclear descriptions have been modified over the revised manuscript.

  1. On line 57, authors mention that GZO has less reactivity in oxygen and lesser moisture resistance compared with other dopants. It would be helpful to elaborate more on how these properties would make GZO interesting for optoelectronics.

It is a typo. We have re-written the description as: GZO has attracted interest in TCO and optoelectronic applications because Ga atom has less reactivity with oxygen and is more moisture resistance as compared with Al atom in page 2 in blue color.

  1. On line 72, more elaboration on the differences in crystalline structure and electrical properties of GZO thin films using different growth technologies will be helpful. Elaboration on applications where these properties are inferior when grown by other technologies will also be helpful.

We have re-written the GZO thin films grown by different technologies as: The PLD and MBE grown-GZO thin films exhibit excellent crystalline structures and low resistivity; nevertheless, scaling them up to standard industrial substrate sizes is difficult [22]. The ALD, sputtering, and thermal oxidation can produce high-quality GZO thin films on a large wafer size; however, these technologies have a low deposition rate that cannot satisfy the demands of mass production. The aqueous solution deposition and the sol–gel method can be used to deposit GZO thin films on large-sized substrates; however, the crystalline structure and electric properties of such GZO thin films grown by these technologies are inferior owing to multiple crystalline orientations and grain boundary scattering. MOCVD grown-GZO thin films show a high crystalline structure, low resistivity, and reasonable growth rate; however, high quality GZO thin films should be deposited in a high growth temperature. The plasma-enhanced MOCVD (PEMOCVD) system, which contained a radio-frequency (RF) power to assist in decomposing precursors, was used to reduce the growth temperature. in page 2 in orange color.

4, Labeling of components in the Figures 1 (a) and 1(b) will enable better understanding of the schematics.

We have labeled the components in the Fig. 1 in page 4.

  1. Caption in Figure 1 (c) is inaccurate and needs to be updated

The caption of figure 1 (c) have be re-written as: (c) the average thickness of MIPEMOCVD- and PEMOCVD-grown GZO thin films at different position on glass substrate in page 4 in red color. The MIPEMOCVD and PEMOCVD are the deposition systems with and without modified intake system. We have added “PEMOCVD systems (without modified intake system)” in page 3 in red color.

  1. The carrier concentration at Ga content of 9% is lower when grown using MIPEMOCVD than PEMOCVD. Elaboration on this trend would be appreciated.

The carrier concentration of MIPEMOCVD-grown GZO is lower than that of PEMOCVD-grown GZO under TEGa flow rate of 9 sccm owing to the strong interstitial mechanism in MIPEMOCVD-grown GZO, which have been shown in page 3 in pink color.

  1. In the modified intake system, the organic precursors are mixed in a tank outside the reaction chamber. For MOCVD, the reactions occur in the reaction chamber under specific growth conditions (temperature, pressure, etc.). How is it made sure that reactions do not occur in the tank outside the reaction chamber before the precursors are sprayed in the reaction chamber? Elaboration on this will be appreciated here and in the manuscript.

The metal-organic precursors will not react in the modified intake system owing to no oxygen free radical and very low deposition pressure. We have added it in page 3 in blue color.

  1. In caption 1 (c), authors mention that the figure represents thickness of GZO thin films. However, it appears that thickness of only one sample is presented in the figure. If this one sample is representative of all other grown samples, please mention so and update the caption.

The description of Figure 1 (c) is ambiguous. We have re-written it as: Fig. 1(c) shows the average thickness of MIPEMOCVD- and PEMOCVD-grown GZO thin films at upper-left (UL), lower-left (LL), center, upper-right (UR), and lower-right (LR) positions of 1.5 cm×1.5 cm glass substrate under the DEZn and TEGa flow rates of 37 and 7 sccm. The standard deviations of thickness for MIPEMOCVD- and PEMOCVD-grown GZO thin films in Fig. 1 (c) are 4.6 and 30.6, implying that the modified intake system can be used to expand the mixed precursors of DEZn and TEGa in the reaction chamber and to obtain a well-controlled deposition rate of MIPEMOCVD-grown GZO thin films over the substrate. in page 3 in orange.

  1. Selection of chamber pressure and deposition temperature for the growth should be justified. Are these optimum conditions for ZnO growth? Please mention the basis for the growth conditions.

We have added table 2 to show the optimal deposition temperature. The optimal chamber pressure can be found in our previous study (ref. 40). The discussion of optimum deposition temperature was written in page 3 in green color.

  1. On line 164, do the authors mean “annealed” instead of “alloyed”? If so, please correct the error.

We mean the formation of alloy between metal and GZO. It may be ambiguous to read. We thanks reviewer`s suggestion and have corrected is as annealed in page 5 in blue color.

  1. On line 168, do the authors mean “emitted” light instead of “emitting light”? If so, please correct the error.

We thank reviewer`s suggestion and have corrected it as emitted in page 5 in orange color.

  1. Please clearly mention the x-axis and y-axis titles on the plots in Figure 3.

The x-axis and y-axis of EDX are X-ray energy and count of elements. We have added it in page 5 in pink color.

  1. Please mention or explain if the increase in Ga content in Figure 3 was proportional (or directly related) to the input TEGa/DEZn ratio.

We have added figure 3 (f) to show relationship between Ga content and TEGa flow rate under the fixed DEZn flow rate.

  1. Line 210 makes an interesting point regarding the incorporation mechanism of Ga in ZnO. It would be great if the sentence could be split into 2-3 sentences or updated to convey the message clearly.

We thank reviewer`s suggestion and have re-written it as: Because Ga atoms were homogeneously incorporated into the ZnO crystal structure, the diffraction angles of (002) peak for MIPEMOCVD-grown GZO thin films with Ga contents of 1.95 and 3.01 at% were close to that of ZnO thin film in the inset of Fig.4. As the Ga contents of MIPEMOCVD-grown GZO thin films rose to 4.52 and 5.42 at%, the diffraction angles of (002) peak shifted to the small angle as compared with that of ZnO thin film in the inset of Fig. 4. The diffraction angle of (002) peak for high Ga substitution at the Zn site in GZO is larger than that for ZnO because the bond length of Zn-O is larger than that of Ga-O. According to the Scherrer formula [41], the crystalline size of GZO will decrease with increasing Ga content owing to the larger diffraction angle. However, MIPEMOCVD-grown GZO thin films with Ga contents of 4.52 and 5.42 at% exhibit a larger calculated grain size and smaller diffraction angles of (002) peak as compared with ZnO thin film (table 3), which are attributed to the stronger interstitial mechanism in MIPEMOCVD-grown GZO thin films with high Ga content [42].in page 6 and 7 in blue color.

  1. Line 238 seems to have an error. The word (high) after deposition seems erroneous. Please correct or clarify the sentence.

The sentence is ambiguous. We have re-written as: The MIPEMOCVD-grown GZO thin films grown under a low deposition pressure and low growth temperature are ordered and directional because the deposition rates of bulk and initial stage are slow and constant for the low deposition pressure but high and slow for the high deposition pressure. in page 8 in orange color.

  1. On line 242, authors mention Al atoms doped in Is this result from this manuscript or from literature? Please clarify.

This result was elaborated in the added ref 45 in page 8 in blue color.

  1. The authors state that reduction in FWHM of (002) peak indicate formation of structural defects (line 248). Please provide more elaboration on this.

It is a typo. We have re-written the paragraph as: Additionally, the FWHM of (002) peak for MIPEMOCVD-grown GZO thin films decreases with increasing the Ga content from 0 to 5.42 at% as shown in Table 3. The MIPEMOCVD-grown GZO thin film with a higher Ga content exhibits a larger grain size that reduces the lattice distortion and structural defects caused by the in-plane stress; as a result, the FWHM of (002) peak for MIPEMOCVD-grown GZO thin film with high Ga content becomes narrow [46] in page 8 in pink color.

  1. Structural characterization (XRD) indicated that Ga occupied more interstitial sites in ZnO (even in GZO with 1.95 and 3.01% Ga). As per line 261, authors indicate that Ga substituted Zn more than occupying interstitial sites. Please explain these observations.

The explanation of Ga substituted Zn is ambiguous. We have re-written it as: The diffraction angles of (002) peak for MIPEMOCVD-grown GZO thin films with Ga contents of 1.95 and 3.01 at% are close to that for ZnO as compared with MIPEMOCVD-grown GZO thin films with Ga contents of 4.52 and 5.42 at%, implying a strong substituted mechanism rather than interstitial mechanism for MIPEMOCVD-grown GZO thin films with Ga contents of 1.95 and 3.01 at%. The ratio of Zn/O is 0.76 and 0.65 for MIPEMOCVD-grown GZO thin films with Ga contents of 1.95 and 3.01 at%, respectively. The 3.01 at% Ga-contented GZO thin film with a lower Zn/O ratio implies more Ga substitution to Zn-site in GZO as comparing to 1.95 at% Ga-contented GZO, in page 9 in blue color.

  1. On line 284, authors mention that the trend of carrier mean free path is comparable to grain size. The grain size increased proportional to the Ga content. But the carrier mean free path does not show such a trend. Please elaborate and update the corresponding sentence(s) in the manuscript.

The sentence to elaborate the increasing carrier mobility with carrier concentration is ambiguous. We have re-written as: The mean free path of carrier was shorter than that of the grain size presented in Table 3; thus, besides ionized impurity scattering, the effect of grain boundary scattering should be considered for the carrier mobility. in page 9 in pink color

  1. In legend of Figure 6, do authors mean “carrier density” instead of “carrier”? If yes, please update legend in Figure 6.

Figure 6 exhibits the hall measurement and carrier concentration is the common use rather than carrier density in literature.

  1. In line 379, authors mention that the p-n heterojunction’s depletion region is “fairly”. Please elaborate on this.

The sentence is ambiguous. We have re-written it as: the width of depletion region in n-ZnO and p-GaN is fairly in page 12 in blue color.

  1. Referring to line 387, please elaborate the effect of joule-heating on the center wavelength of emission spectra. If the effect is not significant, please mention so.

The original description is ambiguous. We have re-written it as: Additionally, the center wavelength of n-ZnO/p-GaN LEDs with GZO shows a red shift possible due to high series resistance. The high series resistance results an increasing junction temperature, leading to a slight reduction of bandgap and red shift in emitting spectrum in page 12 in pink color.

  1. The light output power from GZO/ZnO/GaN lED was more than ITO/ZnO/GaN LED, but the EQE was similar. The authors provide explanation on this around line 334. Please show plots of the EQE. Which optoelectronic applications are influenced more by EQE which are influenced more by light output power characteristics? Please elaborate on this and more on the relevance and application-related impact of results from Figure 8.

We have added the external quantum efficiency in Fig. 8 (a) in page 12. Additionally, the n-ZnO/p-GaN LED with the optimal MIPEMOCVD-grown GZO exhibited a higher light output power than the device with ITO at the same injection current owing to the higher external quantum efficiency (EQE) as shown in Fig. 8 (a). The EQE defined as the ratio of the number of emitting photons to injected electrons is the product of the internal quantum efficiency (IQE) and light extraction efficiency (LEE), which determine the light output power. We have added above description to explain the relation between light output power and external quantum efficiency in page 11 in blue color.

Round 2

Reviewer 2 Report

The manuscript has been improved. The methodologies and findings are better explained in the revised version. However, the manuscript needs some more improvement. Following are the suggestions.

  1. On line 124, the authors mention that the metal-organic precursors will not react in the modified intake system. The sentence especially “no oxygen free radical” is not clear. Also, low deposition pressure could sometimes be conducive for the deposition. Please clarity this.
  2. On line 236, the authors mention that Ga occupies interstitial sites at high Ga content. However, from Figure 3(c), it appears that Ga substituted Zn at high Ga content. Please explain this observation.
  3. In legend (not the title) of Figure 6, do authors mean “carrier concentration (or density)” instead of “carrier”? If yes, please update legend in Figure 6.
  4. The sentence on line 397 “the width of depletion region in n-ZnO and p-GaN is fairly” is ambiguous. Please fix it or find better word or phrase instead of fairly if needed.
  5. Please fix any remaining language-related errors. Following are a few examples.

On line 59, it should probably be “moisture-resistant” instead of “moisture-resistance”. The sentence on line 57 “GZO has attracted….Al atom” probably should be moved to line 48 where there is discussion on Al-doped ZnO.

On line 66, the sentence should probably start with “PLD and MBE” instead of “The PLD and MBE”. On line 68, it should likely be “ALD, sputtering, ….” Instead of “The ALD, sputtering,…”

Author Response

Reply to reviewer’s comments

    We thank the constructive comment made by the reviewer. The specific comments made by the reviewers and replies are listed as following.

  1. On line 124, the authors mention that the metal-organic precursors will not react in the modified intake system. The sentence especially “no oxygen free radical” is not clear. Also, low deposition pressure could sometimes be conducive for the deposition. Please clarity this.

   The description of modified intake system is ambiguous. We have re-written as: because the oxygen entered the reaction chamber directly and the mixed metal-organic precursors will go to the reaction chamber rapidly owing to the low deposition pressure in page 3 in green

  1. On line 236, the authors mention that Ga occupies interstitial sites at high Ga content. However, from Figure 3(c), it appears that Ga substituted Zn at high Ga content. Please explain this observation.

   The “high Ga content” is not clear. We have changed it as: Ga content above 4.52 at% in page 7 and 10 in green.

  1. In legend (not the title) of Figure 6, do authors mean “carrier concentration (or density)” instead of “carrier”? If yes, please update legend in Figure 6.

   We have changed the legend as: Carrier Concen. in page 6

  1. The sentence on line 397 “the width of depletion region in n-ZnO and p-GaN is fairly” is ambiguous. Please fix it or find better word or phrase instead of fairly if needed.

   We thank reviewer’s suggestion and have re-written the sentence as: the width of the depletion region in n-ZnO and p-GaN is almost equal in page 12 in green

  1. Please fix any remaining language-related errors. Following are a few examples.

On line 59, it should probably be “moisture-resistant” instead of “moisture-resistance”. The sentence on line 57 “GZO has attracted….Al atom” probably should be moved to line 48 where there is discussion on Al-doped ZnO.

   We thank reviewer’s suggestion. We have modified the wording errors in blue color over the manuscript and shifted the sentence on line 57 to line 48.

  1. On line 66, the sentence should probably start with “PLD and MBE” instead of “The PLD and MBE”. On line 68, it should likely be “ALD, sputtering, ….” Instead of “The ALD, sputtering,…”

   The wording errors have be changed them to the reviewer’s suggestion.
